# Surgical Management of the Axilla in Invasive Lobular Carcinoma in the Z1071 Era: A Propensity-Score Matched Analysis of the National Cancer Database

Heather F. Sinner [1],*, Samer Naffouje [2] , Julia M. Selfridge [1], Marie C. Lee [1] , Susan J. Hoover [1] and Christine Laronga [1]

[1] Department of Breast Oncology, H. Lee Moffitt Cancer Center, Tampa, FL 33612, USA
[2] Department of Surgical Oncology, H. Lee Moffitt Cancer Center, Tampa, FL 33612, USA
* Correspondence: heather.sinner@caromonthealth.org; Tel.: +1-(813)-745-4673; Fax: +1-(813)-745-7287

**Abstract:** In patients with invasive lobular carcinoma (ILC) and clinically positive nodes (cN1) who demonstrate an axillary clinical response to neoadjuvant-chemotherapy (NAC), the outcomes of sentinel lymph node biopsy (SLNB) compared to axillary lymph node dissection (ALND) are not well studied. We sought to evaluate axillary surgery practice patterns and the resultant impact on overall survival (OS) in cN1 ILC. The National Cancer Database (NCDB) was queried (2012–2017) for women with cN1 ILC who were treated with NAC followed by surgery. Propensity-score matching was performed between SLNB and ALND cohorts. Kaplan–Meier and Cox regression analyses were performed to identify predictors of OS. Of 1390 patients, 1192 were luminal A ILCs (85.8%). 143 patients (10.3%) had a complete axillary clinical response, while 1247 (89.7%) had a partial clinical response in the axilla. Definitive axillary surgery was SLNB in 211 patients (15.2%). Utilization of SLNB for definitive axillary management increased from 8% to 16% during the study period. Among 201 propensity-score matched patients stratified by SLNB vs. ALND, mean OS did not significantly differ (81.6 ± 1.8 vs. 81.4 ± 2.0 months; $p$ = 0.56). Cox regression analysis of the entire cohort demonstrated that increasing age, grade, HER2+ and triple-negative tumors, and partial clinical response were unfavorable OS predictors ($p$ < 0.02 each). The definitive axillary operation and administration of adjuvant axillary radiation did not influence OS. In cN1 ILC patients with a clinical response to NAC in the axilla, SLNB vs. ALND did not affect OS. Further axillary therapy may be warranted with ypN+ disease.

**Keywords:** invasive lobular carcinoma; national cancer database; NCDB; axillary management





## 1. Introduction

Breast cancer is the most commonly diagnosed cancer and leading cause of cancer death for women in the world [1]. Invasive lobular carcinoma (ILC) represents 10–15% of invasive breast cancers, and tends to be diagnosed at a more advanced stage, with larger sized tumors and more frequent nodal involvement than ductal carcinomas [2,3]. ILC is more commonly mammographically occult and less commonly forms a palpable mass, which may contribute to its higher stage at diagnosis [4]. Given its lack of desmoplastic reaction, nodal involvement is also frequently under-diagnosed in the clinical setting [5]. Ten-year survival among persons with ILC as compared to invasive ductal carcinoma (IDC) is lower (86% versus 91%) [6]. A recent meta-analysis demonstrates that patients with ILC are significantly (3-fold) less likely to achieve a pathologic complete response in the breast or axilla following neoadjuvant systemic therapy than their ductal counterparts [7].

Management of the node positive axilla following neoadjuvant chemotherapy in breast cancer is currently debated. Traditional management of the axilla is axillary lymph node dissection for any clinically positive nodes. ACOSOG Z1071 demonstrated that in clinically node positive patients who underwent neoadjuvant chemotherapy, sentinel lymph node

biopsy of at least 2 sentinel nodes with dual-agent mapping yields a false negative rate (FNR) of 12.6%, and 3 sentinel nodes lowers the FNR to 7%, which is considered an acceptable FNR to forego the morbidity of a completion axillary dissection if there was no residual disease in the sentinel nodes [8–10]. The Z1071 trial specifically studied patients who underwent neoadjuvant chemotherapy, but in more recent years, consideration of neoadjuvant systemic therapy (NAST) includes neoadjuvant anti-endocrine therapy (NET); however the universal equivalence of NAC to NET is still being studied [11–13]. Seeing as ILC made up <10% of the sampled population for each of these ACOSOG studies, and no subgroup analysis was conducted, it is unclear if lobular cancers can be held to the same standard as ductal carcinomas. Furthermore, we need to characterize which patients in the ILC population have a pathologic complete response (pCR) in the axilla, to better predict who would benefit from targeted axillary surgery. Does receptor status matter in response to NAST specifically in the ILC population? We sought to evaluate surgical practice patterns and the impact on survival in clinically node positive ILC patients.

## 2. Methods

The National Cancer Database (NCDB) was queried from 2012 to 2017 for this analysis. Women, age 18–90 years, with a diagnosis of unilateral invasive lobular carcinoma on histologic sampling, were selected. All receptor subtypes were included (i.e., any combination of estrogen, progesterone, or Her2Neu receptor status were included). Patients needed to be clinically node-positive (cN1), have documentation of clinical T stage, and have undergone surgery of their breast and axilla. We defined receptor status as Luminal A for estrogen receptor (ER) positive tumors, and Luminal B for ER-negative tumors. Women included underwent NAST, which consisted of any combination of NAC and/or NET. Furthermore, patients were only included if they demonstrated a clinical response to the NAST, which is a defined variable in the NCDB. Women were excluded for de Novo distant metastasis.

After establishment of the appropriate study population, the cohort was divided into SLNB and ALND groups based on the reported definitive axillary surgical management as indexed into the NCDB. Persons with completion ALND or staged SLND followed by ALND were grouped with the ALND cohort. We used conditional logistic regression to compare categorical variables and mixed effect modeling to compare continuous variables between the unmatched groups. A propensity score was calculated based on a logistic regression model that included all other demographic and clinical variables: age, race/ethnicity, Charlson score, breast laterality, receptor status, Nottingham grade, clinical T stage, clinical response (partial or complete), type of mastectomy, axillary management, in-breast response, pathologic nodal status, adjuvant whole-breast or axillary radiation, adjuvant endocrine therapy, and adjuvant systemic therapy. Patients were then matched between the two groups based on a 1:1 ratio following the nearest neighbor method with a 0.1 caliper width, and a mandatory exact match for pathologic nodal status. Kaplan–Meier method was used to study OS, and the log-rank test was used to compare OS outcomes. Identical matching and inferential methodologies were followed in the subsets of patients with ypN0 (pathologically confirmed node negative following NAST) and ypN+ (pathologically confirmed node positive following NAST) to compare OS between SLNB vs. ALND in each subset. Finally, Cox regression analysis was applied in the unmatched cohort of all the patients and in those with ypN+ who had SLNB as a definitive axillary surgical management to identify independent predictors of OS. Statistical significance was set at <0.05 throughout the study. IBM SPSS v25 (Armonk, NY, USA) with R (3.3.3 version) Essentials' plug-in was used to perform data analysis.

## 3. Results

The NCDB between 2012 and 2017 included 1,436,519 new incidences of breast cancer in the United States; 132,169 of these cancers were invasive lobular carcinoma. After applying our selection parameters, as defined in the methods, 1390 adult women were

included. Figure 1 represents a flow diagram of the inclusion/exclusion criteria and the study design.

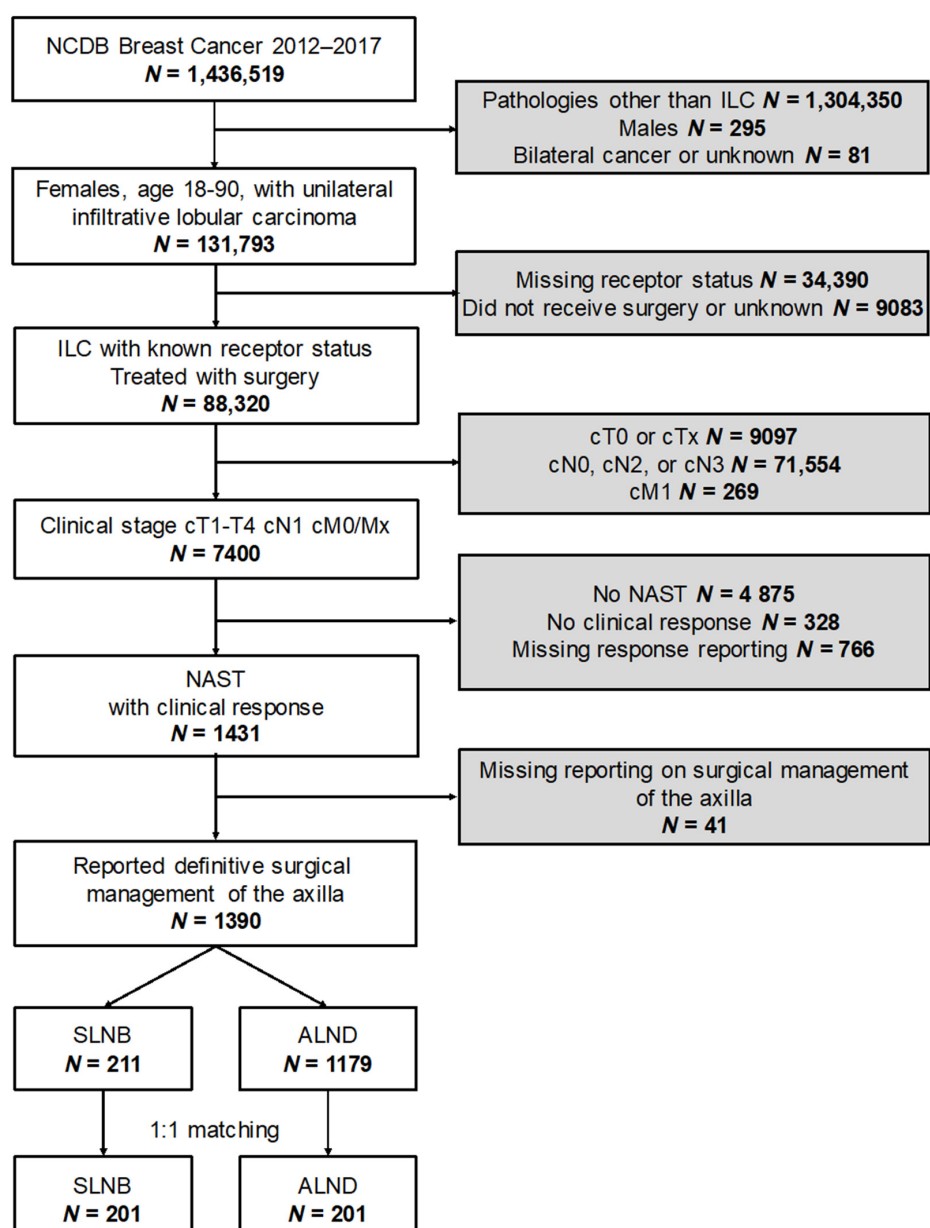

**Figure 1.** Flow diagram summarizing the selection steps and the matching design. ALND: Axillary Lymph Node Dissection; NAST: Neoadjuvant Systemic Therapy; NCDB: National Cancer Database; ILC: Infiltrative Lobular Carcinoma; SLNB: Sentinel Lymph Node Biopsy.

The mean age of the studied population was 58.4 ± 10.3 years (median: 58 years) and majority of patients identified as white race (*N* = 1077; 77.5%). Majority of patients were healthy (Charlson score 0, *N* = 1186, 85.3%), and cancer laterality was equally distributed (left: 51.3%, right 48.7%) (Table 1). Most cancers were Luminal A (*N* = 1192, 85.8%), intermediate grade (*N* = 817, 58.8%), and clinical T2 or T3 disease (cT2: *N* = 475, 34.2%; cT3: *N* = 594, 42.7%). Sixty-five percent of patients underwent mastectomy (*N* = 904) and 49.5% of patients demonstrated at least partial, or complete in-breast response to neoadjuvant chemotherapy (*N* = 687). In the adjuvant setting, 59.5% of patients received adjuvant systemic therapy, 84.5% of patients received endocrine therapy.

**Table 1.** Demographic and perioperative characteristics of the selected cohort ($N$ = 1390). Luminal A: designates an estrogen positive tumor; Luminal B: designates an estrogen negative tumor; ALND: Axillary Lymph Node Dissection; NOS: Not otherwise specified; SD: Standard Deviation; SLNB: Sentinel Lymph Node Biopsy.

| Characteristic | | N (%) |
|---|---|---|
| **Age** | Mean $\pm$ SD (median) | 58.4 $\pm$ 10.3 (58) |
| **Race/Ethnicity** | White | 1077 (77.5%) |
| | Black | 151 (10.9%) |
| | Other | 162 (11.7%) |
| **Charlson Score** | 0 | 1186 (85.3%) |
| | 1 | 166 (11.9%) |
| | 2 | 25 (1.8%) |
| | 3+ | 13 (0.9%) |
| **Laterality** | Right | 677 (48.7%) |
| | Left | 713 (51.3%) |
| **Receptor status** | Luminal A | 1192 (85.8%) |
| | Luminal B | 74 (5.3%) |
| | HR- HER2+ | 55 (4.0%) |
| | Triple negative | 69 (5.0%) |
| **Nottingham grade** | Low | 231 (16.6%) |
| | Intermediate | 817 (58.8%) |
| | High | 170 (12.2%) |
| | Not reported | 172 (12.4%) |
| **Clinical T stage** | cT1 | 153 (11.0%) |
| | cT2 | 475 (34.2%) |
| | cT3 | 594 (42.7%) |
| | cT4 | 168 (12.1%) |
| **Clinical response** | Complete | 143 (10.3%) |
| | Partial | 1247 (89.7%) |
| **Breast surgery** | Lumpectomy | 486 (35.0%) |
| | Mastectomy | 904 (65.0%) |
| **Axillary management** | SLNB | 211 (15.2%) |
| | SLNB-ALND staged | 57 (4.1%) |
| | SLNB-ALND same | 221 (15.9%) |
| | ALND | 901 (64.8%) |
| **In-breast response** | Complete response | 126 (9.1%) |
| | Partial response | 561 (40.4%) |
| | No response | 703 (50.6%) |
| **Pathologic node status** | Negative | 239 (17.2%) |
| | Positive | 1151 (82.8%) |
| **Whole breast radiation** | | 446 (32.1%) |
| **Axillary radiation** | | 690 (49.6%) |
| **Endocrine therapy** | | 1175 (84.5%) |
| **Adjuvant systemic therapy** | | 827 (59.5%) |

Ultimately, majority of patients underwent ALND ($N$= 1179, 84.8%), with 15.9% having SLNB followed by completion ALND at the index operation ($N$ = 221), and 4.1% of patients undergoing ALND at a subsequent operation following final surgical pathology ($N$ = 57). There was an uptrend in the use of SLNB as definitive axillary surgical management, from 8% in 2012 to 16% in 2017 (Figure 2). Residual nodal disease was noted on final pathology in 82.8% of patients (ypN+) ($N$ = 1151, 82.8%).

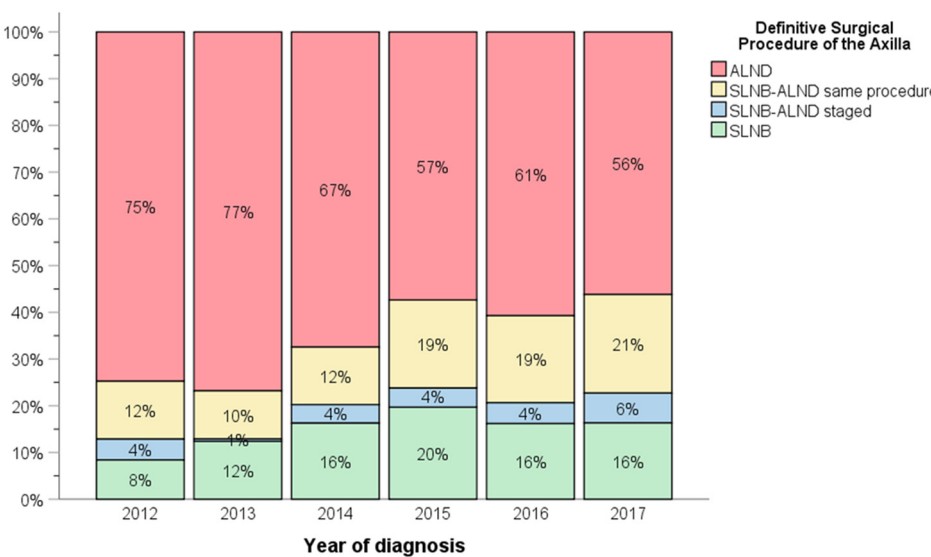

**Figure 2.** Chronological trends of the definitive surgical axillary procedure between 2012–2017.

The cohort was then divided into two groups based on the definitive surgical management of the axilla: SLNB or ALND. Table 2 provides a comparison of the clinical and demographic characteristics of these groups. Baseline comparison of these groups demonstrates that SLNB patients were more likely to be older, have lower clinical T stages, higher rates of in-breast response to NAC (pCR 13.7% vs. 8.2%), lower rates of pathologically node-positive disease, and less likely to undergo axillary radiation ($p < 0.04$ for each). Among 201 propensity-score matched patients stratified by SLNB versus ALNB, all baseline differences resolved in the matched dataset with adequate balance in all the variables. Kaplan–Meier analysis of the matched dataset demonstrates no significant difference in overall survival between SLNB versus ALND (mean OS $\pm$ SD, 81.6 $\pm$ 1.8 vs. 81.4 $\pm$ 2.0 months, $p = 0.052$). Median OS was not reached in either group (Figure 3).

**Table 2.** Comparison of baseline characteristics between the unmatched and matched patients who underwent SLNB vs. ALND as a definitive surgical axillary management. Luminal A: designates an estrogen positive tumor; Luminal B: designates an estrogen negative tumor; ALND: Axillary Lymph Node Dissection; NOS: Not otherwise specified; pCR: Pathologic Complete Response; pNR: Pathologic No Response; pPR: Pathologic Partial Response; SLNB: Sentinel Lymph Node Biopsy. * Statistically significant.

| KERRYPNX | Unmatched Dataset | | | 1:1 Matched Dataset | | |
|---|---|---|---|---|---|---|
| | SLNB | ALND | *p* | SLNB | ALND | *p* |
| **N** | 211 | 1179 | | 201 | 201 | |
| **Age** | 60.0 ± 10.8 | 58.1 ± 10.2 | **0.013 *** | 59.8 ± 10.8 | 59.6 ± 10.7 | 0.886 |
| **Race/Ethnicity** | | | | | | |
| White | 163 (77.3%) | 914 (77.5%) | 0.932 | 158 (78.6%) | 156 (77.6%) | 0.326 |
| Black | 22 (10.4%) | 129 (10.9%) | | 21 (10.4%) | 29 (14.4%) | |
| Other | 26 (12.3%) | 136 (11.5%) | | 22 (10.9%) | 16 (8.0%) | |
| **Charlson Score** | | | | | | |
| 0 | 177 (83.9%) | 1009 (85.6%) | | 169 (84.1%) | 164 (81.6%) | |
| 1 | 30 (14.2%) | 136 (11.5%) | 0.593 | 28 (13.9%) | 34 (16.9%) | 0.836 |
| 2 | 3 (14%) | 22 (1.9%) | | 3 (1.5%) | 2 (1.0%) | |
| 3+ | 1 (0.5%) | 12 (1.0%) | | 1 (0.5%) | 1 (0.5%) | |

**Table 2.** *Cont.*

| KERRYPNX | Unmatched Dataset | | | 1:1 Matched Dataset | | |
|---|---|---|---|---|---|---|
| | SLNB | ALND | *p* | SLNB | ALND | *p* |
| **Receptors** | | | | | | |
| Luminal A | 177 (83.9%) | 1015 (86.1%) | | 169 (84.1%) | 170 (84.6%) | |
| Luminal B | 10 (4.7%) | 64 (5.4%) | 0.493 | 10 (5.0%) | 11 (5.5%) | 0.913 |
| HR- HER2+ | 12 (5.7%) | 43 (3.6%) | | 10 (5.0%) | 11 (5.5%) | |
| Triple negative | 12 (5.7%) | 57 (4.8%) | | 12 (6.0%) | 9 (4.5%) | |
| **Grade** | | | | | | |
| Low | 34 (16.1%) | 197 (16.7%) | | 34 (16.9%) | 31 (15.4%) | |
| Intermediate | 130 (61.6%) | 687 (58.3%) | 0.282 | 121 (60.2%) | 121 (60.2%) | 0.761 |
| High | 29 (13.7%) | 141 (12.0%) | | 28 (13.9%) | 25 (12.4%) | |
| Not reported | 18 (8.5%) | 154 (13.1%) | | 18 (9.0%) | 24 (11.9%) | |
| **Clinical T stage** | | | | | | |
| cT1 | 28 (13.3%) | 125 (10.6%) | | 27 (13.4%) | 18 (9.0%) | |
| cT2 | 80 (37.9%) | 395 (33.5%) | **0.040 *** | 76 (37.8%) | 73 (36.3%) | 0.456 |
| cT3 | 89 (42.2%) | 505 (42.8%) | | 84 (41.8%) | 93 (46.3%) | |
| cT4 | 14 (6.6%) | 154 (13.1%) | | 14 (7.0%) | 17 (8.5%) | |
| **Clinical response** | | | | | | |
| Complete | 29 (13.7%) | 114 (9.7%) | 0.073 | 27 (13.4%) | 33 (16.4%) | 0.401 |
| Partial | 182 (86.3%) | 1065 (90.3%) | | 174 (86.6%) | 168 (83.8%) | |
| **Breast surgery** | | | | | | |
| Lumpectomy | 111 (52.6%) | 375 (31.8%) | **<0.001 *** | 101 (50.2%) | 97 (48.3%) | 0.690 |
| Mastectomy | 100 (47.4%) | 804 (68.2%) | | 100 (49.8%) | 104 (51.7%) | |
| **In-breast response** | | | | | | |
| pCR | 29 (13.7%) | 97 (8.2%) | | 27 (13.4%) | 30 (14.9%) | |
| pPR | 81 (384%) | 480 (40.7%) | **0.037 *** | 75 (37.3%) | 89 (44.3%) | 0.229 |
| pNR | 101 (47.9%) | 602 (51.1%) | | 99 (49.3%) | 82 (40.8%) | |
| **Path node status** | | | | | | |
| Negative | 81 (38.4%) | 158 (13.4%) | **<0.001 *** | 71 (35.3%) | 73 (36.3%) | 0.835 |
| Positive | 130 (61.6%) | 1021 (86.6%) | | 130 (64.7%) | 128 (63.7%) | |
| **Adjuvant chemotherapy** | 124 (58.8%) | 703 (59.6%) | 0.815 | 119 (59.2%) | 108 (53.7%) | 0.268 |
| **Axillary radiation** | 76 (36.0%) | 614 (52.1%) | **<0.001 *** | 76 (37.8%) | 75 (37.3%) | 0.918 |
| **Endocrine therapy** | 177 (83.9%) | 998 (84.6%) | 0.287 | 169 (84.1%) | 166 (82.6%) | 0.688 |

Finally, we performed a Cox univariate and multivariate regression analysis to identify the significant predictors for OS in the unmatched cohort ($N = 1390$). The final multivariate model demonstrated that increasing age, grade, partial clinical response to NAC (versus cCR), and Her2Neu positive or triple-negative tumors (versus Luminal A) were unfavorable predictors of OS ($p < 0.02$ for each) (Table 3). Similarly, we applied a Cox regression analysis to the unmatched subset of patients with ypN+ disease ($N = 1151$) and found similar prognosticators of poor OS: increasing age, partial clinical response (versus cCR), Her2Neu positive or triple-negative tumors (versus Luminal A), and clinical T4 (reference cT1) ($p < 0.03$ for each) (Table 4). The definitive surgical procedure of the axilla (SLNB versus ALND), surgical management of the breast (lumpectomy versus mastectomy), and administration of adjuvant systemic therapy or radiation were not significant predictors in either univariate models.

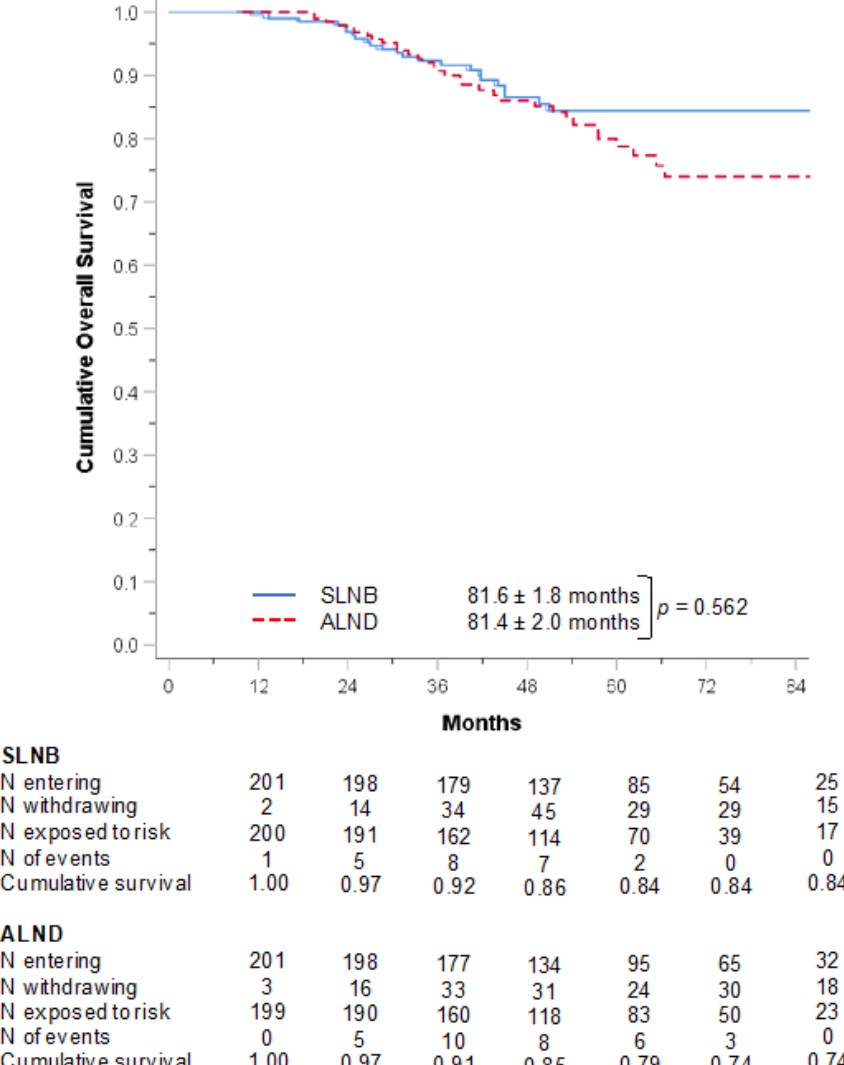

**Figure 3.** Kaplan–Meier analysis of the matched cohort comparing overall survival (mean ± SD) of SLNB vs. ALND as definitive axillary procedures in ILC patients with cN1 who demonstrated clinical response to neoadjuvant systemic chemotherapy. ALND: Axillary Lymph Node Dissection; ILC: Infiltrative Lobular Carcinoma; SD: Standard Deviation; SLNB: Sentinel Lymph Node Biopsy.

**Table 3.** Final step of the backward conditional Cox multivariate regression analysis for predictors of overall survival in the unmatched dataset of all patients (*N* = 1390). CI: Confidence Interval. * Statistically significant.

| | | Hazard Ratio (95% CI) | *p* |
|---|---|---|---|
| **Age** | | 1.020 (1.007–1.033) | **0.002 *** |
| **Grade** | Low | Referent | |
| | Intermediate | 1.881 (1.207–2.930) | **0.005 *** |
| | High | 2.040 (1.170–3.558) | **0.012 *** |
| | Not reported | 1.461 (0.847–2.521) | 0.173 |
| **Receptor status** | Luminal A | Referent | |
| | Luminal B | 0.940 (0.490–1.804) | 0.853 |
| | HR- HER2+ | 2.465 (1.331–4.565) | **0.004 *** |
| | Triple negative | 3.202 (2.091–4.905) | **<0.001 *** |
| **Clinical response** | Complete | Referent | |
| | Partial | 2.496 (1.405–4.433) | **0.002 *** |

**Table 4.** Final step of the backward conditional Cox multivariate regression analysis for predictors of overall survival in the unmatched dataset of ypN+ patients (*N* = 1151). CI: Confidence Interval. * Statistically significant.

|  |  | Hazard Ratio (95% CI) | *p* |
|---|---|---|---|
| **Age** |  | 1.019 (1.005–1.034) | **0.010 *** |
| **Receptor status** | Luminal A | Referent |  |
|  | Luminal B | 0.840 (0.341–2.071) | 0.705 |
|  | HR- HER2+ | 4.616 (2.362–9.020) | **<0.001 *** |
|  | Triple negative | 4.077 (2.496–6.662) | **<0.001 *** |
| **Clinical T stage** | T1 | Referent |  |
|  | T2 | 0.910 (0.530–1.561) | 0.732 |
|  | T3 | 1.380 (0.825–2.308) | 0.221 |
|  | T4 | 1.798 (1.114–3.232) | **0.030 *** |
| **Clinical response** | Complete | Referent |  |
|  | Partial | 2.455 (1.180–5.108) | **0.016 *** |

## 4. Discussion

The extent of surgery in the axilla for breast cancer has been de-escalating over time. Historically, an axillary nodal dissection included axillary lymph node stations I-III, which has evolved to dissection of levels I-II nodes, followed by sentinel node biopsy without axillary node dissection in clinically node negative patients [14,15]. Now, since publication of ACOSOG Z1071, patients with known clinical node positive disease (cN1) having NAC can consider SLNB with potential to forego axillary dissection if pCR [10]. Understandably, the greater majority of patients in these landmark studies had invasive ductal carcinoma. Since it has been studied that overall survival is similar for patients who undergo sentinel node biopsy rather than axillary node dissection in cN1 disease, we analyzed the NCDB in order to assess the trends in use of sentinel node biopsy over axillary dissection in the invasive lobular breast cancer cohort and assessed characteristics of patients who were selected for sentinel node biopsy as definitive axillary nodal surgery.

In this NCDB analysis of clinically node positive ILC patients having neoadjuvant systemic treatment, we confirm that the use of sentinel node biopsy as definitive surgery has increased over the selected period, however, axillary node dissection remains most common. Characteristics of those undergoing sentinel node biopsy as definitive surgery included: older age, lower clinical T stages, higher rates of in-breast response to NAC. However, when directly comparing the SLNB group to the ALND group, in both unmatched and matched cohorts, there was no significant difference in mean overall survival. These results indicate that as long as a patient demonstrates some clinical response to neoadjuvant systemic treatment, then sentinel node biopsy is non-inferior to axillary node dissection and does not change overall survival.

Interestingly, the cohort of patients who underwent SLNB only includes 61.6% (*N* = 130) who despite a positive node at the time of surgery, did not have a completion ALND. Speculation of why this occurred would be related to both patient and provider characteristics. Perhaps the patient refused further surgery. However, perhaps the surgeon extrapolated data from ACOSOG Z0011 and applied those recommendations to patients who had NAST for their locally advanced breast cancer. Perhaps patients included in this database were participants in the Alliance 11,202 trial: a phase 3 non-inferiority multicenter trial ongoing at over 1200 sites examining recurrence and survival outcomes comparing ALND to SLNB and axillary radiation therapy among patients with locally advanced breast cancer. The results of this trial are highly anticipated and have the potential to radically change surgical management of the axilla.

There are several limitations to this study. The standardized variable coding and administrative nature of the database limit the NCDB in analyzing additional granular data points. However, the trade-off is worthwhile in terms of the sheer size of the database

and its reflection of national cancer care patterns at the American College of Surgeons Commission on Cancer accredited hospitals. Despite well over a million breast cancer patients entered in the NCDB, the patients that fit our inclusion criteria are limited by complete data available within the database, especially data on receipt of NAST, which greatly decreased the sample size. Furthermore, to create a propensity-matched dataset, the sample size is even further narrowed. In order to make the sample size more robust, we included patients with both complete and partial response to NAST. Despite a small sample size, it is important to study lobular carcinomas to eliminate biases in practice patterns. As well, the NCDB does not report disease free specific survival, so we are unable to extrapolate the risk of locoregional recurrence in the studied cohort. It is critical to emphasize that patients who demonstrated no response to NAST were excluded from this study, limiting the applicability of results to patients with at least partial response to NAST.

**5. Conclusions**

In cN1 ILC patients who demonstrated a clinical response to NAST in the axilla, the choice of SLNB or ALND did not affect OS, in this limited dataset. Patients that could be considered candidates for targeted dissection are those with lower clinical T stage who demonstrate at least a partial response to NAST.

**Author Contributions:** Study Concept and Design: H.F.S., S.N. and C.L.; Analysis: H.F.S., S.N. and C.L.; Drafting of Manuscript: H.F.S. and C.L.; Revision of Manuscript: H.F.S., J.M.S., M.C.L., S.J.H. and C.L. All authors have read and agreed to the published version of the manuscript.

**Funding:** This research received no external funding.

**Institutional Review Board Statement:** This study is exempt from Institutional Review Board approval as the National Cancer Database is a public use file (PUF) with deidentified data to the extent that persons and providers are no longer a limited data set as defined by HIPAA.

**Informed Consent Statement:** Informed consent was waived due to utilization of a de-identified NCDB file.

**Data Availability Statement:** Data supporting reported results can be publically accessed through written permission from the American College of Surgeons, Commission on Cancer, National Cancer Database. The data used in the study are derived from a de-identified NCDB file. The American College of Surgeons and the Commission on Cancer have not verified and are not responsible for the analytic or statistical methodology employed, or the conclusions drawn from these data by the investigator.

**Acknowledgments:** The authors would like to acknowledge administrative support for conducting this research.

**Conflicts of Interest:** The authors declare no conflict of interest.

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
