# Peer review of "Surgical Management of the Axilla in Invasive Lobular Carcinoma in the Z1071 Era: A Propensity-Score Matched Analysis of the National Cancer Database"

_curroncol, doi:10.3390/curroncol29110647_

Round 1

Reviewer 1 Report

An interesting population to study with - historically - a paucity of data, due to its relatively low incidence compared to IDC.

Line 42 - Management of the node-positive axilla after neoadjuvant chemotherapy is currently debated.

Line 51 - Neoadjuvant endocrine treatment

Line 74 - how can SLND follow ALND? Should this be the opposite way around?

Line 172 - forego axillary dissection if there is an axillary pCR.

Line 186 - caution should be exercised when drawing far-reaching conclusions from these results, due to inherent selection bias. Pts with bad biology disease with poor response and a large residual burden of disease in their SLNs (eg ypN1-2 disease) would be selected into the ALND group, and therefore it is not truly a fact that survival in patients undergoing SLNB in this particular cohort is non-inferior to ALND in this cohort.

Author Response

Line 42 - Management of the node-positive axilla after neoadjuvant chemotherapy is currently debated. Response: Revised sentence as follows for clarity: “Management of the node positive axilla following neoadjuvant chemotherapy in breast cancer is currently debated.”

Line 51 - Neoadjuvant endocrine treatment   Response: We believe the statement is correct, that we do not know if particular molecular subtypes of invasive lobular carcinomas are impacted by either neoadjuvant chemotherapy or neoadjuvant endocrine therapy in alternative manner than ductal carcinomas. This is why we used the language “NAST” (or Neoadjuvant systemic therapy), to be all encompassing.

Line 74 - how can SLND follow ALND? Should this be the opposite way around? Response: Thank you for seeing this important error in verbiage. We have now corrected the sentence as follows, “Persons with completion ALND or staged SLND followed by ALND were grouped with the ALND cohort”

Line 172 - forego axillary dissection if there is an axillary pCR. Response: Exactly, we agree, if the Alliance 11202 trial concludes that survival and recurrence outcomes are similar in setting of SLNB or ALND, then the possibility to forego axillary dissection would greatly decrease surgical morbidity – perhaps even in the setting of only a partial clinical response. We did not edit this statement further because we do not want to make hard conclusions that are not able to be supported by more robust prospective data.

Line 186 - caution should be exercised when drawing far-reaching conclusions from these results, due to inherent selection bias. Pts with bad biology disease with poor response and a large residual burden of disease in their SLNs (eg ypN1-2 disease) would be selected into the ALND group, and therefore it is not truly a fact that survival in patients undergoing SLNB in this particular cohort is non-inferior to ALND in this cohort.   Response: We specifically chose a 1:1 propensity match analysis to limit the selection bias when drawing conclusions about the outcomes of SLNB and ALND. However, we agree that your critique is fair and we have adjusted our conclusion statement needs to be less aggressive, given the retrospective nature of our study. We revised the conclusion to state the following, “In cN1 ILC patients who demonstrated a clinical response to NAST in the axilla, the choice of SLNB or ALND did not affect OS, in this limited dataset. Patients that could be considered candidates for targeted dissection are those with lower clinical T stage who demonstrate at least a partial response to NAST.”

Regarding areas of the manuscript that reviewers felt must be improved: We have corrected an error in the methodology that was likely very confusing as to our grouping of patients and feel that figure 1 is extremely important for the reader to understand the step-by-step patient selection process. It is our hope that Figure 1 will be printed in close proximity to the methodology section, for ease of reference. Within our results, our tables appear to have shifted rows, which likely makes readability difficult. We have inserted new versions of the tables into the revised manuscript that have more defined borders, headers, and simplified language. We have adjusted our conclusions to be more appropriate for the level of evidence provided by a retrospective study. As well, we have elaborated on the weaknesses of this study within the discussion section.

Reviewer 2 Report

It is an interesting topic as the management of the axilla in invasive lobular cancer still needs to be studied. ACOSOG Z1071 demonstrated that for neoadjuvant chemotherapy treated cancers is necessary to use dual mapping technique and excise at least 3 sentinel lymph nodes in order to decrease the false negative rate. However for invasive lobular cancer the number of positive lymph nodes can be larger even in luminal A tumour types, as stated by Narbe U & all (U Narbe, P-O Bendahl, M Fernö, C Ingvar, L Dihge, L Rydén, St Gallen 2019 guidelines understage the axilla in lobular breast cancer: a population-based study, British Journal of Surgery, Volume 108, Issue 12, December 2021, Pages 1465–1473, https://doi.org/10.1093/bjs/znab327). 

The article is well written, with strong statistical analysis. 

However the quality of tables and figures must be improved. 

Also is will be interesting to know other parameters of evolution such as recurrence of the disease, and also how many of the patients that didn't undergo further axillary surgery after positive SLN received axillary radiotherapy. 

Author Response

The article is well written, with strong statistical analysis. Response: Thank you, we appreciate your feedback.

However the quality of tables and figures must be improved.  Response: We believe our tables and figures were not easily read and as such, altered the formatting and headings to provide clarity.

Also is will be interesting to know other parameters of evolution such as recurrence of the disease, and also how many of the patients that didn't undergo further axillary surgery after positive SLN received axillary radiotherapy.   – Response: We agree that recurrence data would be of value. Unfortunately, the NCDB does not provide that information to researchers due to concerns for accuracy and lack of completeness.  We have made a statement in the discussion, within the limitations paragraph, addressing this paucity of data.

Regarding areas of the manuscript that reviewers felt must be improved: We have corrected an error in the methodology that was likely very confusing as to our grouping of patients and feel that figure 1 is extremely important for the reader to understand the step-by-step patient selection process. It is our hope that Figure 1 will be printed in close proximity to the methodology section, for ease of reference. Within our results, our tables appear to have shifted rows, which likely makes readability difficult. We have inserted new versions of the tables that have more defined borders, headers, and simplified language. We have adjusted our conclusions to be more appropriate for the level of evidence provided by a retrospective study. As well, we have elaborated on the weaknesses of this study within the discussion section.